# Cell-Free Supernatant of *Bacillus subtilis* Reduces Kiwifruit Rot Caused by *Botryosphaeria dothidea* through Inducing Oxidative Stress in the Pathogen

**DOI:** 10.3390/jof9010127

**Published:** 2023-01-16

**Authors:** Yezhen Fan, Kui Liu, Ruoxi Lu, Jieyu Gao, Wu Song, Hongyan Zhu, Xiaofeng Tang, Yongsheng Liu, Min Miao

**Affiliations:** 1School of Food and Biological Engineering, Hefei University of Technology, Hefei 230036, China; 2Institute of Botany, The Chinese Academy of Sciences, Beijing 230094, China; 3Ministry of Education Key Laboratory for Bio-Resource and Eco-Environment, State Key Laboratory of Hydraulics and Mountain River Engineering, College of Life Science, Sichuan University, Chengdu 610064, China; 4School of Horticulture, Anhui Agricultural University, Hefei 230036, China

**Keywords:** biocontrol, *Botryosphaeria dothidea*, postharvest disease, oxidative damage, kiwifruit

## Abstract

Biological control of postharvest diseases has been proven to be an effective alternative to chemical control. As an environmentally friendly biocontrol agent, *Bacillus subtilis* has been widely applied. This study explores its application in kiwifruit soft rot and reveals the corresponding mechanisms. Treatment with cell-free supernatant (CFS) of *Bacillus subtilis* BS-1 significantly inhibits the mycelial growth of the pathogen *Botryosphaeria dothidea* and attenuates the pathogenicity on kiwifruit in a concentration-dependent manner. In particular, mycelial growth diameter was only 21% of the control after 3 days of treatment with 5% CFS. CFS caused swelling and breakage of the hyphae of *B. dothidea* observed by scanning electron microscopy, resulting in the leakage of nucleic acid and soluble protein and the loss of ergosterol content. Further analysis demonstrated that CFS significantly induces the expression of *Nox* genes associated with reactive oxygen species (ROS) production by 1.9–2.7-fold, leading to a considerable accumulation of ROS in cells and causing mycelial cell death. Our findings demonstrate that the biocontrol effect of *B. subtilis* BS-1 CFS on *B. dothidea* is realized by inducing oxidative damage to the mycelia cell.

## 1. Introduction

Kiwifruit is one of the most resourceful, nutritious, and economically healthy fruits. The freshly ripened kiwifruit is juicy, unique in flavor, rich in vitamins and minerals, and deeply accepted by the market [1,2,3,4]. However, as commercial kiwifruit cultivation and production continue to expand, various diseases have emerged during the harvest and storage of kiwifruit and have become increasingly prominent and severe. These serious diseases include soft rot, grey mold, and black spot disease [5,6,7]. Among them, the soft rot of kiwifruit is the dominant postharvest disease. Despite that there is no difference between the appearance of a diseased fruit and that of a healthy one, the inside of the diseased fruit turns dirty brown and decays, losing edible value. The occurrence of soft rot seriously affects the quality and yield of kiwifruit and is a common cause of major economic losses worldwide [8,9]. Numerous scholars have been working on isolating and identifying the causal agent of the soft rot of kiwifruit. Several major pathogens have been reported to be associated with postharvest rot of kiwifruit, including *Botryosphaeria dothidea*, *Diaporthe actinidiae*, *Phomopsis sp.,* and *Alternaria alternata* [10,11,12,13]. On balance, *Botryosphaeria dothidea* is considered one of the primary pathogens of kiwifruit soft rot worldwide [14]. *B. dothidea,* a necrotrophic pathogen responsible for diseases in a broad range of plant hosts, belongs to the class *Dothideomycetes* of the phylum *Ascomycota* and is distributed worldwide. It is a killer of woody plant species, causing branches’ death, ulcers, gum flow, and postharvest decay of fruits. Kiwifruits infected by *B. dothidea* show a water-soaked lesion spot on the peeled flesh with a light brown center and a dark green ring edge [15,16].

Although certain sorts of chemicals are effective against these pathogens, mixtures of pyraclostrobin and boscalid could change the morphology of conidia and hyphae branches, resulting in suppressed growth of *B. dothidea.* However, these chemical fungicide residues are potentially hazardous to human health and the environment [12,17]. As public concern about the quality and safety of agricultural products rises, researchers are devoted to exploring efficient biological control methods to manage plant diseases. Microbial antagonists against plant pathogens provide an environmentally friendly approach and meet the urgent demand for controlling kiwifruit soft rot diseases [18,19]. *Bacillus spp.* is one of the bacteria broadly distributed in the soil rhizosphere and is widely regarded as a powerful biological control agent against a wide range of pathogenic fungi on a variety of fruits. Volatile organic compounds released by *Bacillus subtilis* strain CL2 inhibited the growth of four pathogenic fungi, contributing to the reduction of postharvest disease in wolfberry [20]. The culture suspension of *B. subtilis* L1-21 effectively inhibited the growth of *Botrytis cinerea* by up to 86.57%, resulting in effective control of postharvest tomato grey mold [21]. In addition, many countries have approved *B. subtilis* as a food supplement, and the toxicity of *B. subtilis* has been evaluated by acute and chronic doses in guinea pigs and rabbits, showing no harmfulness in animals under tested conditions [22]. Although *B. subtilis* is a broad-spectrum antagonistic bacterium suited for disease control in various fruits and vegetables, most works focused on the biological control effect. Moreover, the molecular regulatory mechanism of *B. subtilis* that destroyed pathogens and inhibited their growth is poorly understood. In addition, the inhibition mechanism of *B. subtilis* on kiwifruit soft rot, especially on the primary pathogen *B. dothidea*, has not been extensively studied.

In this study, the biocontrol application of *B. subtilis* was extended to prevent and control the soft rot of kiwifruit. We mainly aimed to examine the inhibitory effect of CFS on soft rot caused by *B. dothidea* in vitro and in vivo and further explore the damage mechanism of CFS on *B. dothidea*. This work will assist in providing a practical, sustainable, and safe biological approach to managing soft rot in kiwifruit.

## 2. Materials and Methods

### 2.1. Microbial Materials

The soft rot fungal pathogen *Botryosphaeria dothidea* was isolated and identified from the perspective of morphology, molecular biology, and Koch’s rule and shares 100% identity with the ITS information of *B. dothidea* strain CMW8000 (AY236949.1). The *B. dothidea* was stored in a −80 °C refrigerator and cultured in potato dextrose agar medium (PDA) at 25 °C.

*Bacillus subtilis* strain BS-1 was kindly provided by Professor Liu Jia from Chongqing College of Arts and Sciences. Its morphologic properties, growth characteristics, and DNA analysis results met the criteria for identifying *B. subtilis*. The 16sRNA sequence of strain BS-1 was 99% identical to that of *B. subtilis* strain CNPMS22169 (MH358457.1).

### 2.2. Preparation of the Cell-Free Supernatant (CFS)

*B. subtilis* strain BS-1 was activated in an LB plate at 28 °C for 16–24 h, and individual clones were transferred to sterile test tubes containing 2 mL LB medium and cultured until the cell cultures OD_600_ reached 0.6 as a seed broth. Fermentation cultures were expanded at 1% of the seed broth inoculum for 72 h and centrifuged at 8000 rpm for 15 min. The supernatant was purified by passing through a 0.22 µm sterile extractor to give a cell-free supernatant (CFS). The CFS was diluted with distilled water to 1%, 2%, and 5% of the final working volume fraction.

### 2.3. Fruits

Mature kiwifruits (*Actinidia deliciosa* cv. Xuxiang) were purchased from supermarkets during the marketing season from September to October with the characteristics of similar size, without decay or physical damage, 8–10% soluble solids content (SSC), and 30–35 N hardness. Experimental kiwifruits were disinfected with 1% (*v/v*) sodium hypochlorite for 10 min, rinsed three times with water, and air-dried at room temperature [23]. 

### 2.4. Effectiveness of the CFS against Kiwifruit Soft Rot Pathogen In Vitro and In Vivo

Mycelial discs (5 mm diameter) from one-week-old cultured *B. dothidea* were placed in the center of PDA plates, amended with CFS at each working volume fraction (1%, 2%, and 5% *v/v*) or without CFS, and further incubated at 25 °C. The inhibition rate was calculated according to the equation [24], and the assay was performed 3 times with 5 replicates. The experimental kiwifruits were divided into five groups and soaked in different concentrations of CFS for 5 min. Punched holes about 2 mm in diameter and 3 mm in depth were created in the equatorial part of the treated kiwifruit with a sterile inoculation spike and inoculated with mycelial discs. The inoculation holes were wrapped in sealing film, and the kiwifruits were incubated under 22–25 °C and 80–90% humidity. The area of rotten spots was calculated with the cross method after 3 and 5 days of affection. The experiment was replicated three times with 12 fruits per treatment group.

### 2.5. Ultrastructural Morphology of Pathogens Treated with the CFS

Mycelial discs (5 mm diameter) from one-week-old cultures of *B. dothidea* were added to culture flasks containing 30 mL of PDB medium and incubated for 48 h at 28 °C on a shaker at 120 rpm. Then, CFS with final volume fractions of 0%, 1%, 2%, and 5% was added to the culture. Mycelia were taken at 0 h, 24 h, and 48 h, then washed three times with 0.05 M PBS (pH 7.2) buffer, fixed with 2.5% (*v/v*) glutaraldehyde (Solarbio, Beijing, China) for 24 h and then dehydrated with graded ethanol (30%, 50%, 70%, and 90%, *v/v*) and freeze-dried as described. The structural features of the mycelium were then observed using a scanning electron microscope (SEM) (Hitachi S-4800, JAPAN) [25].

### 2.6. Effect of the CFS on Cell Death of Pathogenic Mycelium

Fresh mycelium samples treated in step 2.4 were stained using the trypan blue staining kit (Sangon Biotech, Shanghai, China) and observed with a light microscope. Each group was treated with three samples and the experiment was repeated three times.

### 2.7. Determination of the Integrity of Cell Membranes

A few mycelia treated in step 2.4 were incubated with 5 mM propidium iodide (PI) for 20 min at 30 °C and washed twice with 0.05 M pH = 7.4 PBS buffer to remove the stain. Each processing sample was poked and dispersed, then observed with a laser confocal microscope (ZEISS LSM710, Germany). The green light was excited at 405 ± 10 nm, and the red light was absorbed at 633 ± 10 nm [26].

### 2.8. Determination of Ergosterol, Determination of the Leakage of Cellular Contents 

Fresh mycelium samples treated in step 2.4 were taken and rinsed twice with 0.05 M PBS (pH 7.2) buffer to remove residual media. Each sample was placed in a drying oven at 80 °C, dried to a constant mass, ground to a powder with liquid nitrogen, weighed at 0.5 g of the dry powder in a test tube with anhydrous ethanol (1:10, m/V), mixed thoroughly for 10 min, and centrifuged at 3500 rpm for 10 min. The supernatant was filtered through a 0.22 μm microporous membrane, and the ergosterol content was determined at 292 nm [27]. Each group was treated with three samples, and the experiment was repeated three times. 

Nucleic acid and protein exudation were determined by reference to the methods of Bradford et al. with appropriate modifications [28]. Mycelial discs (5 mm diameter) of one-week-old *B. dothidea* were added to shake flasks containing 30 mL of PDB medium and incubated at 25 °C at 120 rpm. After 48 h of incubation, the mycelium was combined and washed with sterile distilled water. Subsequently, 0.5 g of mycelium was suspended in CFS solutions at final volume fractions of 0%, 2%, and 5% and further incubated for 24 and 48 h at 25 °C in a shaker at 120 rpm. After filtering the mycelial samples, the resulting filtrate of the culture was assayed with NANODROP 2000 to determine the concentration of nucleic acids and soluble proteins. Each group was treated with three samples, and the experiment was repeated three times. 

### 2.9. The CFS induces Malondialdehyde (MDA) Production and Reactive Oxygen Species (ROS) Accumulation

The MDA content was determined using the thiobarbituric acid (TBA) method [18]. 2′,7′-dichlorodihydrofluorescein diacetate (H2DCFDA; Invitrogen, Eugene, OR, USA) was used to detect ROS accumulation. Mycelium samples treated in step 2.4 were resuspended in 0.05 M PBS (pH 7.2) and co-incubated for 20 min with 10 μM H2DCFDA, then washed three times with PBS to remove the dye. The ROS accumulation was observed using a confocal laser microscope (LSM800; Zeiss, Oberkochen, Germany) [29]. Each group was treated with three samples, and the experiment was repeated three times. Hydrogen peroxide (H_2_O_2_) content and superoxide anion (O_2_**^·^**^−^) production rate were individually determined using commercial kits (Nanjing Jiancheng Institute of Biological Engineering, Nanjing, China).

### 2.10. Enzyme Activity Assay and Real-Time Fluorescence Quantitative PCR Assay

SOD, POD, and CAT enzyme activities were determined using commercial kits (Jiancheng Institute of Biological Engineering, Nanjing, China). Mycelium samples were collected at 0 h and 6 h according to the method for culturing pathogenic mycelium in step 2.4 and stored at 4 °C and −80 °C until further processing. The total RNA was extracted from mycelium using an RNA Extraction Reagent (Vazyme, R401-01, China), and the cDNA was obtained using a cDNA Synthesis kit (Vazyme, R323-01, China). Real-time qRT-PCR was performed in 10 μL reactions using ChamQTM SYBR^®^ qPCR Master Mix (Vazyme, China) in a fluorescence quantitative PCR instrument (Bio-Rad CFX96). The PCR reaction conditions were as follows: one cycle at 95 °C with initial denaturation for 30 s; then 40 cycles of 15 s at 95 °C and 30 s at 60 °C; and a final cycle of 95 °C at 25 s, 60 °C at 60 s, and 95 °C at 15 s. The gene expression level was normalized by the internal reference gene *Actin* (WWBZ02000009.1) with the 2^−ΔΔCt^ method [30]. The specific primers for the genes evaluated are listed in Table 1. Each treatment consisted of three biological replicates, and the experiments were performed twice.

### 2.11. Effect of N-Acetylcysteine (NAC) on the Recovery of Mycelial Growth

#### 2.11.1. Plate Recovery Experiments

Mycelial discs (5 mm diameter) of one-week-old *B. dothidea* cultures were placed in the center of the PDA discs, with one group containing final volume fractions of 0%, 2%, and 5% CFS and 10 mM NAC. The control group had an identical final volume fraction of CFS and equal amounts of sterile water. Colony diameter was measured after 5 days of inoculation [31]. 

#### 2.11.2. Mycelial ROS Accumulation and Mycelial Cell Damage/Integrity Recovery Test 

Mycelial discs (5 mm diameter) from one-week-old cultures of *B. dothidea* were added to shake flasks containing 30 mL of PDB medium and incubated at 25 °C at 120 rpm. After 2 days of incubation, the cultures were divided into two groups. The treatment group contained different fractions of CFS (0%, 2%, and 5%) and 10 mM NAC, while the control group contained the exact final volume fraction of CFS alone. The two group cultures were incubated at 28 °C and 120 rpm for 24 h. The incubation of mycelium was followed by ROS accumulation staining and the trypan blue staining described previously. 

### 2.12. Statistical Analysis

Statistical analyses were performed using SPSS 23.0, plotted using GraphPad Prism8, analyzed using a one-way analysis of variance (ANOVA), and shown as standard errors of the mean ± least squares mean (SEM). A *p*-value < 0.05 was accepted as statistically significant according to Duncan’s multiple polar difference test. Significant differences are indicated by letters, with different letters indicating significant differences. 

## 3. Result

### 3.1. The CFS of Bacillus Subtilis BS-1 Inhibits Vegetative Growth and Pathogenicity of Botryosphaeria Dothidea 

Mycelial discs (5 mm in diameter) of uniformly sized one-week-old *B. dothidea* cultures were inoculated on PDA plates containing different concentrations of BS-1 CFS to investigate the vegetative growth of *B. dothidea* affected by CFS. After 3 days of incubation, *B. dothidea* growth was suppressed by CFS in a concentration-dependent manner, with mycelial growth at about 66%, 26%, and 21% of the control’s colony diameter under exposure to 1%, 2%, and 5% of CFS, respectively (Figure 1A,B). The growth inhibition properties of *B. dothidea* on the fifth day, similarly to that on the culture on day 3, were affected by the dosage mode of CFS. CFS-treated fruits with the inoculation of mycelial discs were stored at 25 °C to explore CFS’s effects on the virulence of *B. dothidea.* Fruit with CFS treatment developed a substantially and significantly less rotten area than control fruit in a dose-dependent way (Figure 1C,D). Consequently, the optimal suppression was obtained with the 5% CFS treatment; the lesion area was reduced by more than half compared to the control group (Figure 1C,D). *B. subtilis* CFS vitality assays demonstrated dose-response inhibitory effects against *B. dothidea* in vitro and in vivo, showing a promising biocontrol result.

### 3.2. The CFS Damages and Ruptures the Hyphae of B. dothidea

The normal hyphal morphology and the integrity of hyphal structure are necessary for the growth and pathogenicity of *B. dothidea*. The morphological effects of CFS on growing hyphae of *B. dothidea* were examined by microscopy. In the absence of CFS, hyphae exhibited ordinary status over a 24 h incubation period. Compared with the control group, CFS strongly induced striking changes in the hyphal morphology, depending on the CFS concentration. The changes observed included the increasing formation of branches and swelling of hyphal cells (Figure 2A). Based on light microscopic observation, 2% CFS and 5% CFS had a pronounced effect on hyphae. The effects of 1% CFS on pathogenicity inhibition and hyphae morphology were not significantly different compared with the control (Figure 2A); the main focus was explored through the two concentrations of 2% and 5% CFS in further analysis. Then, the scanning electron microscopic analysis further demonstrated that the hyphae in the CFS treatment group were rough and shriveled, in contrast to the plump and smooth hyphae in the control group (Figure 2B). The scanning electron microscopic observation showed that the cell inclusions almost disappeared, but part of collapsed and shriveled hyphae was left (Figure 2B). These results indicated that the CFS primarily affected the normal formation and structure of growing hyphae, and the swelling hyphae cells were accompanied by excessive lateral branching and hyphae fracture that ultimately led to the destruction of the whole hyphae.

### 3.3. The CFS Destroys the Hyphal Cytomembrane and Cause Hyphae death of B. dothidea In Vitro

Shriveled hyphae were observed in the CFS treatment group, suggesting that CFS may damage the cell membrane of hyphae and cause hyphal death. To test this hypothesis, propidium iodide (PI) and trypan blue were employed to detect the membrane integrity of *B. dothidea* cells since the dead cells with incomplete cell membranes could be stained in red or blue by propidium iodide or trypan blue staining, respectively. As shown in Figure 3A, hyphae in the treatment group showed stronger red fluorescence, and 2% of CFS was sufficient to cause severe damage to the cell membranes of hyphae. These results were repeated on the same batch of samples following trypan blue staining. Similar results were obtained in this assay, and the CFS-treated hyphae showed a darker blue than the control (Figure 3B), further proving that CFS retains the ability to destroy cell membranes. 

Cell membrane damage is often accompanied by changes in membrane substances, including the malondialdehyde (MDA), a product of membrane lipid peroxidation, and ergosterol, an essential component of fungal cell membranes. The content of MDA and ergosterol in hyphae revealed major changes between CFS-free and CFS-treated samples. The significantly increased MDA content was detected upon CFS presence at volume fractions of 2% and 5% (Figure 3C). Accordingly, ergosterol decreased significantly after CFS treatment and showed a dose- and time-dependent effect (Figure 3D). The peroxidation-damaged cell membrane may lead to the exudation of cell contents, mainly protein and nucleic acid. Therefore, the concentrations of the above two substances in the hyphal culture medium with or without CFS were further detected. Figure 3E,F show that the protein and nucleic acid content increased conspicuously depending on CFS concentration. These results indicate that the CFS destroyed the integrity of the hyphal membrane of *B. dothidea* and caused cell death by promoting the realization of hyphae membrane peroxidation damage.

### 3.4. The CFS Promotes the Explosion of Reactive Oxygen Species in Hyphal Cells of B. dothidea

In the staining image of CFS-treated hyphae of *B. dothidea*, an incomplete cell membrane was observed. Primary cellular contents of hyphae were also found in the culture medium with CFS. These results lead us to the possible mechanism that CFS is active against fungal cell membrane integrity, probably due to provoking reactive oxygen species (ROS) burst. The cell-permeable fluorescence probe H2DCFDA was applied to detect intracellular ROS in hyphae. After 24 h of treatment with 2% CFS, most of the hyphae gave ROS green fluorescence (Figure 4A). The same fluorescence signal accumulation was also observed in samples treated with a higher concentration of CFS, and such fluorescence increased in extent with an increasing CFS amount. The proliferation of ROS in hyphae triggered by CFS ultimately results in peroxidation of the cell membrane and collapse of the whole hyphae. ROS are chemically active free radical molecules, which could be as simple as molecules of superoxide or more complex molecules, such as hydrogen peroxide. The detection of superoxide and hydrogen peroxide are significant indicators of reactive oxygen species accumulation. As shown in Figure 4B, the mycelium treated with CFS had a significantly higher hydrogen peroxide content and superoxide anion production rate than the control mycelium. Moreover, the production rate depends on the increase in CFS concentration. This result indicates that CFS accelerated the rate of reactive oxygen species production and increased the accumulation of reactive oxygen species. In addition to ROS burst, upregulation of essential ROS synthesis genes (*Nox1*, *Nox1f*, *Nox3*) was observed universally at the very early phase of CFS processing, and the expression level was positively correlated with the concentration of CFS (Figure 4C). These results indicated that the ROS level raised by CFS might be closely related to the damage of hyphae.

### 3.5. Defense-Related Enzyme Activity and Corresponding Defense Enzyme Synthesis-Related Gene Expression in B. dothidea Mycelial Cells

The accumulation of ROS in cells is usually accompanied by the enhancement of scavenging capacity. The enzymatic scavenging system of ROS is largely dependent on the sequential cooperation of superoxide dismutase (SOD), catalase (CAT), and peroxidase (POD), which eventually catalyze ROS into harmless substances. Enzymatic activity was assessed in the hyphae with and without CFS after 24 h of culture. Indeed, a significant increase in scavenging enzyme activity was seen at diverse degrees (Figure 5A). The most pronounced increase in CAT and POD enzyme activity was observed in the group with 5% CFS application, about three times that of the control group. Moreover, the maximal activities of SOD among different CFS treatment groups exhibited similar behavior. The expression levels of genes encoding these ROS-scavenging enzymes were also assessed by qRT-PCR in the same samples. As Figure 5B shows, enzyme synthesis genes (*BdSOD*, *BdPOD*, *BdCAT*) were significantly upregulated at the early stage of CFS treatment. In conclusion, these results suggested that CFS induced ROS rapid accumulation in mycelial cells and leads to mycelial oxidative damage.

### 3.6. N-Acetylcysteine Alleviated the Burst of Reactive Oxygen Species in Hyphae of B. dothidea Induced by CFS

To quantify the oxidative accumulation of hyphae induced by CFS, an inhibition assay was performed. The broad-spectrum ROS inhibitor N-acetylcysteine (NAC) was resuspended in *B. dothidea* hyphae culture with CFS to evaluate the ROS fluorescent signal changes between the CFS-alone group and the CFS and NAC group. As shown in Figure 6, NAC effectively decreased the CFS-induced green fluorescence signal, indicating that NAC could remove various types of ROS generated by CFS. These data proved again from another perspective that CFS of *B. subtilis* induced a considerable accumulation of ROS in hyphae cells, leading to oxidative damage of hyphae.

### 3.7. N-Acetylcysteine (NAC) Reduces Cell Death and Recovers the Weak Growth of B. dothidea Caused by CFS

Considering the scavenging ability of NAC in terms of ROS, we next assessed the ability of NAC to restore the viability and growth of CFS-treated mycelium. Therefore, along with the original pathogen in vitro test, trypan blue staining test, and ROS accumulation staining test, NAC (10 mM) was applied to the CFS treatment groups to observe the status of NAC on hyphal cell death and mycelial growth. Hyphae swelling caused by CFS was hardly observed after NAC addition (Figure 7A), demonstrating that the changes in hyphal morphology induced by CFS were alleviated. In addition to the effect on hyphal morphology rescue, NAC was also effective in alleviating the hyphae death caused by CFS; the images clearly showed that the blue color of hyphae in NAC was much lighter than that of the CFS-treated group (Figure 7A). Referring to the NAC function on recovery hyphae growth under oxidative stress, we measured the CFS-damaged *B. dothidea* mycelia expansion capacity on the PDA plates with or without NAC. Figure 7B,C illustrate the image of the mycelia growth feature and statics of mycelia diameter after treatment with NAC. Regardless of the CFS concentration leveling the plate, the mycelia expansion became faster and more significant with NAC treatment, meaning NAC relieved the inhibition of CFS on mycelial growth. Taken together, these results suggested that the ROS scavenger NAC could rescue the growth disorder of mycelia treated with CFS by removing intracellular ROS.

## 4. Discussion

Fruit is an essential part of the diet, providing people with a rich source of vitamins and minerals, but postharvest losses of fruit amount to around 30% globally [32]. While biological control of postharvest fruit diseases has been subject to extensive attention and research, including the application of biocontrol microorganisms and natural substances from plants, this safe and effective control method remains a hot research topic. *Bacillus halotolerans* KLBC XJ-5 inhibits the growth of *Botrytis cinerea* mycelium and germination of conidia, effectively controlling postharvest strawberry grey mold [33]. Magnolol, a functional component of Magnolia officinalis, activates the autophagic activity of *B. cinerea* and substantially inhibits the mycelial growth of *B. cinerea* [23]. However, a great majority of postharvest biocontrol studies have concentrated on grey mold caused by the model fungus *B.cinerea*, with little attention on other diseases. In addition, studies on the biocontrol of pathogens have mainly focused on the effectiveness of control, but have seldom revealed their inhibition mechanism. These necessitate improvements to the research on the biocontrol of postharvest diseases.

Kiwifruit is a widely received fruit with a high vitamin C content and a rich nutritional profile. Nonetheless, the economic performance of kiwifruit is seriously affected by the infestation of several pathogens, with the soft rot of kiwifruit caused by *Botryosphaeria dothidea* being one of the leading causes affecting fruit production and quality [34]. This demonstrates the urgent need for effective, safe, and efficient disease control measures against the pathogen. Therefore, in this study, we preliminarily found that the *B. subtilis* strain BS-1 had a significant inhibitory effect on kiwifruit soft rot, and the interaction between BS-1 and the pathogen *B. dothidea* was further investigated to assess its potential biotic mechanisms against kiwifruit soft rot. Our results were similar to those of *B. subtilis* in primary postharvest diseases of ecological fruits such as strawberry, grapes, tomato, and cucumber, where treatment with the fermentation solution (CFS) of *B. subtilis* inhibited the expansion of the pathogen in vitro and weakened the pathogenicity of pathogens on the host in a dose-dependent manner [35,36,37,38]. These indicate that *B. subtilis* has a broad-spectrum inhibitory effect and is effective against kiwifruit soft rot. Previous studies have indicated three main mechanisms of biocontrol bacteria: competition for space and nutrients, production of inhibitory substances (cell-free supernatants, CFS and volatile substances), and induction of systemic resistance [39]. Considerable research suggests that inhibition substances releasing is the primary biocontrol method to reduce disease in postharvest fruit and vegetables [40,41]. The CFS of *B. subtilis* EA-CB0015 impedes the development of grey mold by inhibiting conidial germination, shoot tube growth, sporangial formation and tissue colonization of *Botrytis cinerea* [42]. The antifungal action of *B. subtilis* may be attributed to the ability to synthesize a wide range of bioactive secondary metabolites in CFS and volatile components against pathogens [43]. The secondary lipopeptide metabolites of *Bacillus amyloliquefaciens* S76-3 reduce the pathogenicity of *Fusarium graminearum* by disrupting mycelial cell structure and inhibiting mycelial growth and conidial germination [44]. Lipopeptide iturin A extracted from the CFS of *Bacillus subtilis* WL-2 induces cell membrane damage, oxidative stress, and mitochondrial dysfunction of Phytophthora infestans, leading to hyphal cell death [39]. Additionally, several investigations prove that the CFS of *B. subtilis* is highly resistant to a variety of stresses, including high temperature, UV irradiation and low temperature et al. [45,46]. Our previous study demonstrates that the CFS of *B. subtilis* strain BS-1 is tolerant to these stress conditions. Therefore, this study focused on the mechanism of CFS of BS-1 in the biocontrol of kiwifruit soft rot induced by *B. dothidea.*

The results we obtained showed that CFS inhibited the growth of *B. dothidea* in a dosage-dependent manner and effectively suppressed the development of soft rot in kiwifruit in both in vitro and antagonistic inhibition assays (Figure 1). Similarly, our data showed that the growth inhibition of *B. dothidea* was related to the morphological structure of the hyphae, which was damaged by the *B. subtilis* BS-1 CFS treatment, becoming rough, crumpled, and severely fractured (Figure 2B). Based on the observations, we hypothesized that the membrane permeability and intracellular contents of the treated hyphae cells were altered. In this work, the hypotheses were tentatively determined based on propidium iodide (PI) and trypan blue staining, the determination of malondialdehyde (MDA) and ergosterol, and the leakage of cellular contents (Figure 3A–F). It turned out that CFS could disrupt cellular membranes, leading to the leakage of nucleic acids and protein macromolecules from the cells and the alteration of cell permeability (Figure 3A–F). Changing the permeability of the hyphae cell membrane could conduct cell apoptosis or necrosis [43]. Therefore, our results suggest that CFS causes cell death in *B.dothidea* by interfering with the integrity of the cell membrane. 

Reactive oxygen species (ROS) are one of the main factors that induce cellular damage and are a significant marker of oxidative stress [47]. When large amounts of ROS are produced, these will constantly attack and damage cellular tissues, exacerbating peroxidation of the plasma membrane, and leading to damage of the cellular membrane system, thereby destroying the structural and functional integrity of cellular tissues [48,49]. In our study, after 24 h of exposure to CFS, the hyphae ROS fluorescence signal became more robust with increasing CFS concentration. Hydrogen peroxide levels and superoxide anion production rates were significantly higher than those in controls, and ROS synthesis-associated genes (*BdNox1*, *BdNox1f*, *BdNox3*) were upregulated quickly after CFS treatment (Figure 4A–C). The NADPH oxidases (Nox) are *the* most prominent enzyme family with *the* primary function of highly regulated production of ROS, which involves *Nox* genes varying among fungi species [50]. Several biocontrol agents and biological metabolites have been identified as effective elicitors of *Nox* family genes, leading to the accumulation of ROS in pathogens and achieving the control effect. Treatment with *Bacillus velezensis* A4 CFS could upregulate the expression of *BcNoxA* and *BcNoxB* genes and stimulate the accumulation of ROS in the mycelium of *B. cinerea*, inducing oxidative damage and apoptosis in the mycelium [51]. The gene expression of the Nox complex subunit, *BcNoxB*, *BcNoxD*, and *BcNoxR* in the mycelia of *B. cinerea* was upregulated by methyl tartrate, and ROS in the cytoplasm of the mycelium were accumulated, eventually leading to mycelia death [52]. By comparing the homology of *BdNox1*, *BdNox1f,* and *BdNox3* with nox in *B. cinerea*, the homology of *BcnoxA* with *BdNox1* and *BdNox1f* is close to 70.75% and 67.45%, respectively, and the homology of *BcnoxB* with *BdNox3* is close to 66.48%. This suggests that the three *Nox* genes in *B. dothidea* perform functions similar to those genes of *noxA* and *noxB* in *B. cinerea*, as evidenced by our experimental results.

In order to scavenge excess ROS and reduce cell damage, the enzymes associated with ROS elimination (SOD, POD, CAT) and corresponding genes are induced and activated [17]. A 5% BS-1 CFS treatment for 6 h could significantly induce the expression of tested genes (*BdSOD*, *BdPOD*, *BdCAT*), while 2% CFS exhibited a mild induction effect. However, both 2% and 5% CFS resulted in significant increases in the activities of three enzymes in hyphae cells after 24 h of treatment (Figure 5). These results suggest that 2% CFS needs a relatively long time to induce gene expression, or CFS can induce the co-expression of several genes related to enzyme synthesis, which enhances the enzyme activity after CFS treatment at diverse amounts. N-acetylcysteine(NAC), an inhibitor of flavoenzymes such as NADPH oxidase, was introduced for further confirmation [53]; 10 mM NAC partially restored the growth of CFS-induced mycelium of *B. dothidea* and significantly weakened the accumulation of ROS and apoptosis of mycelial cells (Figure 6 and Figure 7). Combined with the MDA content in the CFS condition (Figure 3C) [54], all the above results suggest that CFS of *B. subtilis* BS-1 can directly induce oxidative stress through crucial genes and disrupt the integrity of cell membranes, leading to cell collapse and death and reducing the pathogenicity of *B. dothidea*.

In addition, 1% CFS showed a positive inhibition on the pathogen in vitro and on kiwifruit soft rot in vivo (Figure 1), yet it could hardly affect the morphology and structure of the hyphae (Figure 2). These data implied that CFS of *B.subtilis* BS-1 could inhibit the activity of pathogens more than ROS-induced hyphal damage. Recent works have shown that *B. subtilis* MBI600 are able to stimulate the plant-induced systemic resistance (ISR) signal pathway in the plant, consequently inducing the disease resistance potential of plants [55]. In some cases, *B. subtilis* can reduce the growth of pathogens through nutritional competition, as *B. subtilis* (10-4, 26D) and *B. subtilis* KLBC BS6 [56,57]. The possible mechanism of CFS hampering soft rot caused by *B. dothidea* in our work might be the result of multiple causes. Further work is necessary to identify other potential pathways of CFS inhibition of *B. dothidea* to complete the panorama of the mechanism.

## 5. Conclusions

In conclusion, the cell-free supernatant (CFS) of *Bacillus subtilis* BS-1 exhibited a remarkable inhibitory effect on *B. dothidea* both in vitro and on kiwifruit. By inducing the accumulation of excessive ROS, CFS damaged the membrane of the hyphae cells, eventually leading to the death of hyphae cells, which threatened the normal growth and pathogenicity of *B. dothidea.* These results provide a theoretical basis for the subsequent biological application of *B. subtilis* and biocontrol of kiwifruit soft rot.

## Figures and Tables

**Figure 1 jof-09-00127-f001:**
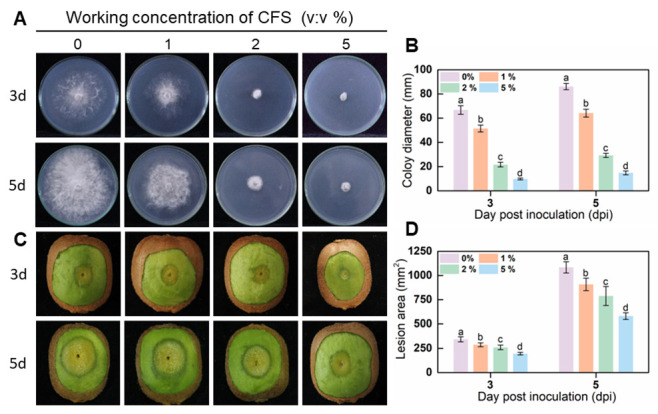
The CFS inhibited vegetative growth of *B. dothidea* on the PDA (potato dextrose agar) plate (**A**,**B**) and reduced the pathogenicity of *B. dothidea* in kiwifruit (**C**,**D**). (**A**) Images were recorded after 3 and 5 days of incubation. (**B**) Statistical analysis for colony diameters. (**C**) Photographs of diseased fruits were taken 3 and 5 days after treatment. (**D**) Lesion area on harvested kiwifruit was measured using Image J. Histograms with different letters indicate significant differences according to Duncan’s multiple range tests at *p* < 0.05.

**Figure 2 jof-09-00127-f002:**
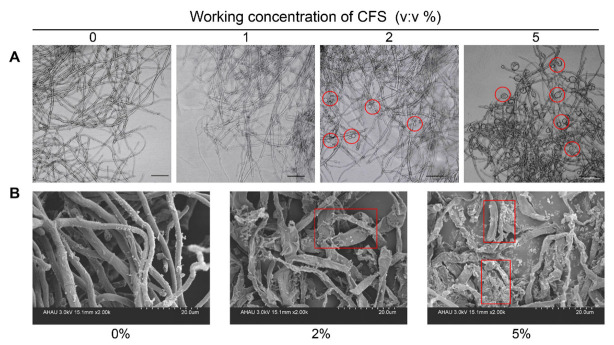
Effects of CFS on the morphological structure of *B. dothidea*. (**A**) Effects of CFS on the morphological structure of *B. dothidea* were tested in a PDB medium containing the indicated concentration of CFS. The hyphae were collected after 24 h of treatment, and images were recorded using a microscope. Black bars represent 50 μm. Hyphae swollen into spheres are marked by red circles. (**B**) Scanning electron micrographs of *B. dothidea* treated with CFS at different working concentrations. The shriveled hyphae are marked by red rectangles.

**Figure 3 jof-09-00127-f003:**
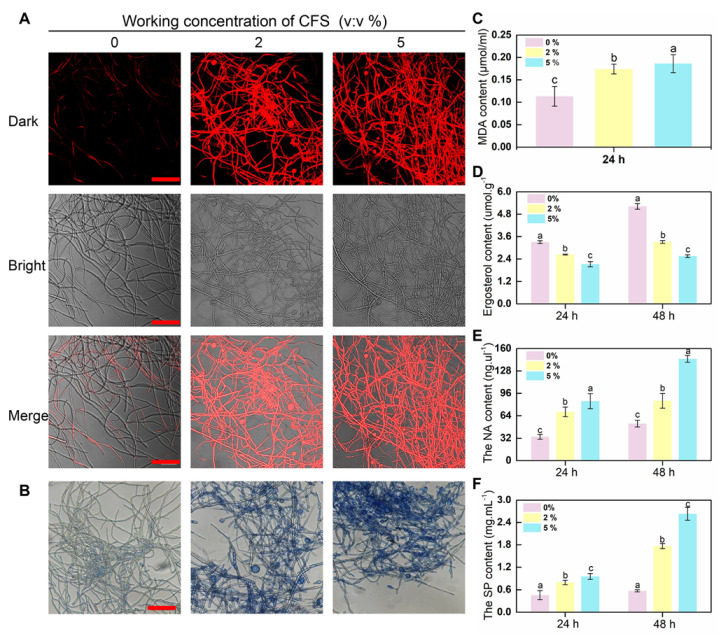
Injury efficacy of CFS on the hyphal cytomembrane of *B. dothidea*. (**A**) Detecting the integrity of the membrane of hyphal cells stained by propidium iodide (PI). The fluorescence signal was detected using a laser scanning confocal microscope at 405 nm excitation and 633 nm emission. (**B**) Detecting the integrity of membrane and activity of hyphal cells at 24 h after the application of CFS stained by trypan blue. The images were recorded using a microscope. The red bars represent 50 μm. (**C**,**D**) The content of MDA and ergosterol in hyphae after treatment with CFS for 24 h and 48 h. (**E**,**F**) Nucleic acid (NA) and Soluble protein (SP) contents were measured in a hyphal culture medium after different amounts of CFS t reatment at indicated period. Vertical bars represent standard deviations of the means. Columns followed by different letters are statistically different by the Duncan’s multiple ranges (*p* < 0.05).

**Figure 4 jof-09-00127-f004:**
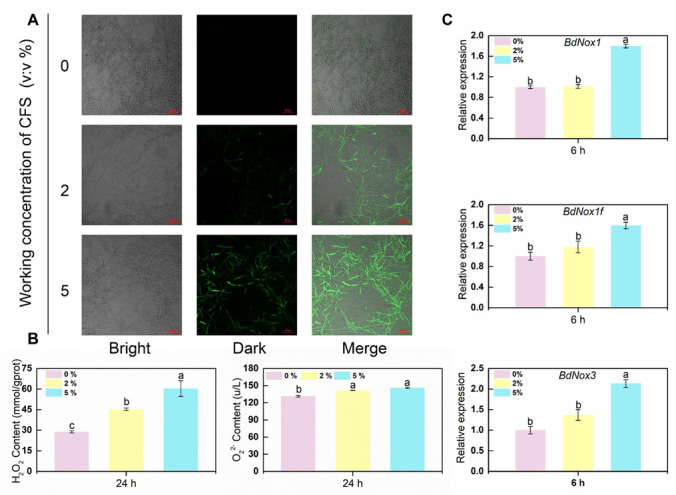
Effects of CFS on reactive oxygen species (ROS) accumulation in *B. dothidea*. (**A**) The images were recorded using confocal laser microscopy at 488 nm excitation and 520 nm emission. The hyphal cells treated with different concentrations of CFS were harvested after 24 h. Harvested hyphal cells were stained with the oxidant-sensitive probe, H2DCFDA, and then washed with PBS buffer for observation. (**B**) The contents of H2O2 and O_2_**^·^**^−^ measured after 24 h treatment. (**C**) Gene expression levels of BdNox1, BdNox1f, and BdNox3 at 6 h after treatment. Different letters above the columns indicate significant differences within each group according to Duncan’s multiple range test (*p* < 0.05).

**Figure 5 jof-09-00127-f005:**
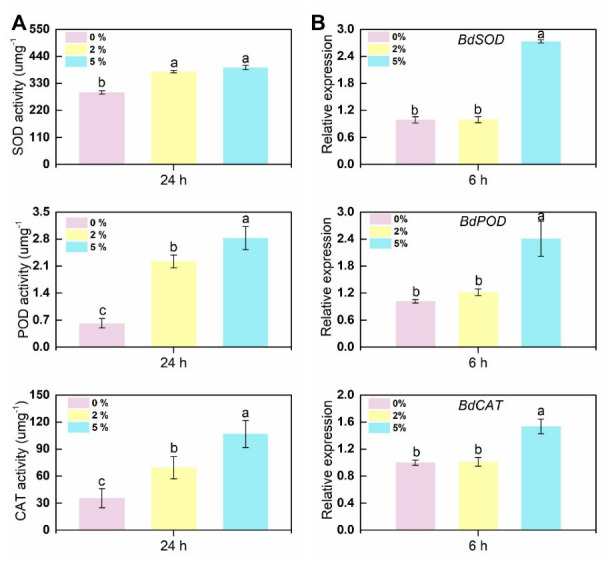
Effects of CFS on the enzyme activities involved in scavenging reactive oxygen species in *B. dothidea*. (**A**) The activities of ROS scavenging enzymes (POD, SOD, and CAT) were recorded in harvested hyphal cells. (**B**) Gene expression levels of *BdSOD*, *BdPOD,* and *BdCAT* at 6 h after treatment. Different letters above the columns indicate significant differences within each group according to Duncan’s multiple range test (*p* < 0.05).

**Figure 6 jof-09-00127-f006:**
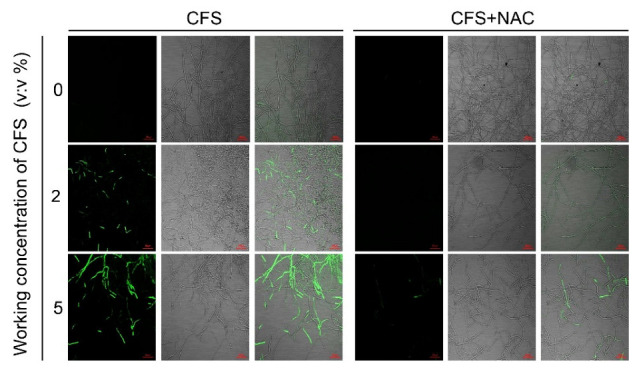
Effect of N-acetylcysteine (NAC) on the accumulation of reactive oxygen species (ROS) in hyphae. The culture medium was amended with different concentrations of CFS with or without 10 mM NAC, and the representative fluorescent pictures were recorded using confocal laser microscopy at 488 nm excitation and 520 nm emission.

**Figure 7 jof-09-00127-f007:**
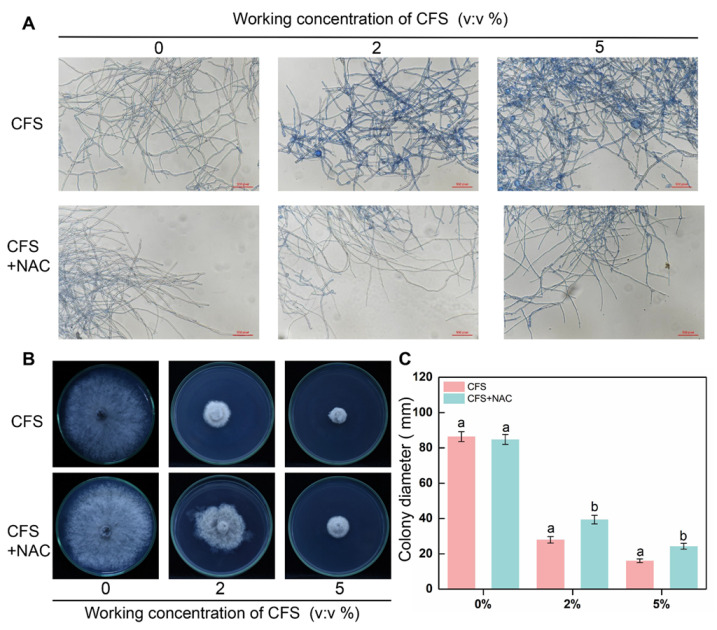
Effect of N-Acetylcysteine (NAC) on the activity and growth of mycelia treated with CFS. The cultures were amended with CFS (2% and 5% *v/v*) and 10 mM NAC, separately. (**A**) Hyphal cell death was determined by an accumulation of staining with trypan blue after the application of CFS and treatment with NAC for 24 h. Images were recorded using a microscope. (**B**) The mycelial growth diameter was measured, and images were recorded 5 days after inoculation. (**C**) Statistical analysis for colony diameters. Error bars indicate standard errors of the means of two repeated experiments. Treatments annotated with different letters were significantly different within a specific time point according to the Duncan’s multiple range test. (*p* < 0.05).

**Table 1 jof-09-00127-t001:** Primers Used for the RT-qPCR Analysis.

Gene Name	Accession Number	Primer Name	Primer Sequences (5′→3′)
*Nox1*	WWBZ02000033.1	*BdNox1-F1*	CGAGTCGATATGGTTCCACAG
		*BdNox1-R1*	GCCTGGGTATAGTGAATCTGG
*Nox1f*	WWBZ02000073.1	*BdNox1f-F1*	TGGCATCTACCTTTTCGAGC
		*BdNox1f-R1*	CCAGGCTTGTACTTCATCGAG
*Nox3*	WWBZ02000020.1	*BdNox3-F1*	CGTTTCCAAGGTTATTCAGCAC
		*BdNox3-R1*	TTGTGAGAGTGAAAGGGTGG
*Sod*	WWBZ02000033.1	*BdSod-F1*	GTCGGTGACAACTCTGGC
		*BdSod-R1*	GGTAATGGATCACGATGCTG
*Pod*	WWBZ02000001.1	*BdPod-F1*	TCTCGGCTACAAGATCGGAG
		*BdPod-R1*	TCACAGAATCCGGCAAGC
*Cat*	WWBZ02000022.1	*BdCat-F1*	TATTCCTCAGGCACGATCTTG
		*BdCat-R1*	GTCTATTGAGAAGGGTCGGTTC
*Actin*	WWBZ02000009.1	*BdActin-F1*	GGTTCAACTACCACCTCAAGAATG
		*BdActin-R1*	GCCGTGGGCGTCAGAAAT

## Data Availability

Not applicable.

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
