# Peer review of "Cell-Free Supernatant of Bacillus subtilis Reduces Kiwifruit Rot Caused by Botryosphaeria dothidea through Inducing Oxidative Stress in the Pathogen"

_jof, 2023, doi:10.3390/jof9010127_

Round 1

Reviewer 1 Report

The article “Cell-free supernatant of Bacillus subtilis reduces kiwifruit rot caused by Botryosphaeria dothidea through inducing oxidative stress in the pathogen” by Fan et al., demonstrated the biocontrol action mechanism of Bacillus subtilis against Botryosphaeria dothidea in kiwifruit. Through series of microscopic and molecular and biochemical investigations, various mechanisms such as cell membrane damage by inducing oxidative stress in the pathogen, changing the cell permeability, leaking internal contents out of the cell etc. were confirmed. The study is interesting and conducted in enough depth to be published in JoF. However, before its publication it needs major revisions in writing or presentation of the manuscript. Consider the following pints in this regard.

Keywords should not be in title

Almost half of the abstract describes the background of the study. Followed by some results in descriptive form and a speculation at the end. There is a need to rewrite the whole abstract. Give a concise background in 2-3 lines followed by giving results in more accurate form (by giving some digits) and a concluding statement at the last instead of any speculation. The abstract in this way will provide an effective overview of the study.

Instead of giving findings at the end of introduction part, objectives or hypothesis should be explained.

Introduction section is lacking the information about pathogen

Introduction section needs to be elaborated in particular the authors should highlight the novelty of this work, and particular illustrate the superiority of this work from previous reports. Also, what are the gaps in this regards.

Microbes used in this study should be identified properly on molecular basis. No information was provided in this regard.

The title of first paragraph in material and methods should be revised as in this paragraph information about the preparation of CFS was also given

It is important to mention the date/month of purchasing fruit

The mixture of reaction volume used in quantitative PCR and operating conditions are not given. What is used for internal reference? and how the quantitative expression was measured?

The manuscript should be carefully revised so that the results are better discussed. All sections should be critically discussed and compared with the previous reports. This will actually strengthen the manuscript and will highlight the significance of the study.

In all sections of the Result chapter, the last concluding statements should be placed and discussed in discussion chapter

It may be remembered that this Conclusions Section forms a summary of all the major observations/ results obtained. Accordingly, here presentation should consist of the main Results or the observations of the study in short sentences. This should stand alone or form a subsection of a Discussion or Results Section. Hence better to rewrite this Section and condense this section.

Authors are encouraged to add some up to date literature references for discussion of the manuscript as many new reports are available from the year 2018-2022.

References should be revised carefully especially for scientific names as majority are not in italics. Moreover, follow the JoF style for writing journal name and over all reference

Author Response

Dear Editor,

Enclosed please find the revised manuscript according to the valuable comments. We are very grateful for all the comments regarding our manuscript. We have done a considerable amount of editing to address their concerns, all changes are marked as red. We hope that these amendments are sufficient to warrant publication in JoF.

Best regards

Min Miao

Point to point responses:

Comments of Reviewer 1:

Comment 1: Keywords should not be in title

Answer:Thanks for your comments, and we have modified in the revised manuscript (line 28).

Comment 2: Almost half of the abstract describes the background of the study. Followed by some results in descriptive form and a speculation at the end. There is a need to rewrite the whole abstract. Give a concise background in 2-3 lines followed by giving results in more accurate form (by giving some digits) and a concluding statement at the last instead of any speculation. The abstract in this way will provide an effective overview of the study.

Answer:Thanks for your valuable suggestions. We have accordingly rewritten the abstract (lines 14-26).

Comment 3: Instead of giving findings at the end of introduction part, objectives or hypothesis should be explained.

Answer:Thanks for your comments. We have revised this part according to your suggestion (lines 75-79).

Comment 4: Introduction section is lacking the information about pathogen

Answer: Thanks for the advice. We have accordingly added the information about pathogen Botryosphaeria dothidea in the revised manuscript. (lines 46-51).

Comment 5: Introduction section needs to be elaborated in particular the authors should highlight the novelty of this work, and particular illustrate the superiority of this work from previous reports. Also, what are the gaps in this regard.

Answer:Thanks for your comments. We have revised this part according to your suggestion (lines 69-74).

Comment 6: Microbes used in this study should be identified properly on molecular basis. No information was provided in this regard.

Answer:Thanks, and we have provided the relevant information in the revised manuscript (lines 89-91).

Comment 7: The title of first paragraph in material and methods should be revised as in this paragraph information about the preparation of CFS was also given

Answer:The mentioned title was revised according to your suggestion (line 92).

Comment 8: It is important to mention the date/month of purchasing fruit

Answer:Thanks for the comments. We have included the purchase date information in the revised manuscript (line 102). In addition, the biocontrol experiment on kiwifruit was accomplished within one week after the fruit was purchased, and the experiment was repeated three times with similar results.

Comment 9:  The mixture of reaction volume used in quantitative PCR and operating conditions are not given. What is used for internal reference? and how the quantitative expression was measured?

Answer: We apologize for our brief description of quantitative RT-PCR, and all your concerned details have been added in the revised manuscript (lines 175-183).

Comment 10: The manuscript should be carefully revised so that the results are better discussed. All sections should be critically discussed and compared with the previous reports. This will actually strengthen the manuscript and will highlight the significance of the study.

Answer: Thanks for your valuable comments. We have made a comprehension discussion and rewrote this section.

Comment 11: In all sections of the Result chapter, the last concluding statements should be placed and discussed in discussion chapter

Answer: Thanks, we have made modifications according to your suggestions.

Comment 12: It may be remembered that this Conclusions Section forms a summary of all the major observations/ results obtained. Accordingly, here presentation should consist of the main Results or the observations of the study in short sentences. This should stand alone or form a subsection of a Discussion or Results Section. Hence better to rewrite this Section and condense this section.

Answer: Thanks for your comments. The conclusion section has been rewritten and condensed according to your suggestion (lines 508-513).

Comment 13: Authors are encouraged to add some up to date literature references for discussion of the manuscript as many new reports are available from the year 2018-2022.

Answer: Thanks for your comments. We have included some recent literature in the discussion section.

Comment 14: References should be revised carefully especially for scientific names as majority are not in italics. Moreover, follow the JoF style for writing journal name and over all reference

Answer: We apologize for our incorrect format, and we have carefully revised all references according to JoF style.

Reviewer 2 Report

The manuscript titled "Cell-free supernatant of Bacillus subtilis reduces kiwifruit rot caused by Botryosphaeria dothidea through inducing oxidative stress in the pathogen" is overall good and interesting, although the idea is old. Meanwhile, some sections of the manuscript, particularly the material and methods section, need to be rewritten. The English language needs improvement. Also, the manuscript had many spelling errors that needed to be revised. The discussion part was poorly written and must be rewritten.

Regards

Author Response

Dear Editor,

Enclosed please find the revised manuscript according to the valuable comments. We are very grateful for all the comments regarding our manuscript. We have done a considerable amount of editing to address their concerns, all changes are marked as red. We hope that these amendments are sufficient to warrant publication in JoF.

Best regards

Min Miao

Point to point responses:

Comments of Reviewer 2:

Comment: The manuscript titled "Cell-free supernatant of Bacillus subtilis reduces kiwifruit rot caused by Botryosphaeria dothidea through inducing oxidative stress in the pathogen" is overall good and interesting, although the idea is old. Meanwhile, some sections of the manuscript, particularly the material and methods section, need to be rewritten. The English language needs improvement. Also, the manuscript had many spelling errors that needed to be revised. The discussion part was poorly written and must be rewritten.

Answer: Thanks for the valuable comments. We have rewritten the materials and methods section and discussion part and carefully corrected spelling errors. We have asked a native English speaker to polish our paper.

Round 2

Reviewer 1 Report

Manuscript has been revised according to the suggested comments. 

Author Response

Comment 1:Manuscript has been revised according to the suggested comments. 

Answer: Thanks for the valuable comments. We have made numerous corrections, including grammar, spelling, and presentation. All changes are marked up using the “Track Changes” function.

Reviewer 2 Report

I think that the manuscript could be accepted after English correction 

Author Response

Comment 1: I think that the manuscript could be accepted after English correction.

Answer: Thanks for the valuable comments. We have made numerous corrections, including grammar, spelling, and presentation. All changes are marked up using the “Track Changes” function.